# A Comprehensive Review on the Role of Non-Coding RNAs in the Pathophysiology of Bipolar Disorder

**DOI:** 10.3390/ijms22105156

**Published:** 2021-05-13

**Authors:** Soudeh Ghafouri-Fard, Elham Badrlou, Mohammad Taheri, Kenneth M. Dürsteler, Annette Beatrix Brühl, Dena Sadeghi-Bahmani, Serge Brand

**Affiliations:** 1Department of Medical Genetics, School of Medicine, Shahid Beheshti University of Medical Sciences, Tehran 65167198721, Iran; s.ghafourifard@sbmu.ac.ir (S.G.-F.); elhambadr95@gmail.com (E.B.); 2Skull Base Research Center, Loghman Hakim Hospital, Shahid Beheshti University of Medical Sciences, Tehran 65167198721, Iran; 3Psychiatric Clinics, Division of Substance Use Disorders, University of Basel, 4002 Basel, Switzerland; Kenneth.Duersteler@upk.ch; 4Center for Addictive Disorders, Department of Psychiatry, Psychotherapy and Psychosomatics, Psychiatric Hospital, University of Zurich, 8001 Zurich, Switzerland; 5Center for Affective, Stress and Sleep Disorders (ZASS), Psychiatric University Hospital Basel, 4002 Basel, Switzerland; annette.bruehl@upk.ch (A.B.B.); dena.sadeghibahmani@upk.ch (D.S.-B.); 6Sleep Disorders Research Center, Kermanshah University of Medical Sciences, Kermanshah 67146, Iran; 7Departments of Physical Therapy, University of Alabama at Birmingham, Birmingham, AL 35294, USA; 8Department of Sport, Exercise and Health, Division of Sport Science and Psychosocial Health, University of Basel, 4052 Basel, Switzerland; 9School of Medicine, Tehran University of Medical Sciences, Tehran 25529, Iran

**Keywords:** bipolar disorder, circRNA, miRNA, lncRNA

## Abstract

Aim: Bipolar disorder is a multifactorial disorder being linked with dysregulation of several genes. Among the recently acknowledged factors in the pathophysiology of bipolar disorder are non-coding RNAs (ncRNAs). Methods: We searched PubMed and Google Scholar databases to find studies that assessed the expression profile of miRNAs, lncRNAs and circRNAs in bipolar disorder. Results: Dysregulated ncRNAs in bipolar patients have been enriched in several neuron-related pathways such as GABAergic and glutamatergic synapses, morphine addiction pathway and redox modulation. Conclusion: Altered expression of these transcripts in bipolar disorder provides clues for identification of the pathogenesis of this disorder and design of targeted therapies for the treatment of patients.

## 1. Introduction

Bipolar disorder (BD) is a multifactorial disorder characterized by the occurrence of severe mood impairment episodes, neuropsychological complications, immunological alterations, and perturbation in personal/social functions [1]. As one of the main sources of disability all over the world [2], BD is associated with premature death from the co-existence of other medical conditions as well as suicide attempts [3,4]. Several genetic and environmental parameters have been recognized to modulate the risk of BD, yet most of them being liked with a number of other mental disorders as well. The causal link between a few of these risk factors and BD has been established [5]. Among the recently acknowledged factors in the pathophysiology of BD are non-coding RNAs (ncRNAs) [6]. These transcripts participate in the epigenetic marking of several genes through modulating chromatin configuration and RNA editing. Their binding with complementary sequences in the genome might alter methylation or RNA sites. Moreover, long ncRNAs (lncRNAs) are implicated in the complicated regulatory systems that control the expression of target genes [7]. Transcript profiling in autopsy samples of medial frontal gyrus from bipolar patients and non-psychiatric controls have shown differential expression of ten lncRNA transcripts and a global higher number of alternative spliced variants in these patients [6]. Other types of ncRNAs such as microRNAs (miRNAs) and circular RNAs (circRNAs) have also been dysregulated in brain tissues or peripheral blood of patients with BD [6,8]. We performed a comprehensive search in PubMed and Google Scholar databases to find studies that assessed expression profile of miRNAs, lncRNAs and circRNAs in BD. This study is study is a narrative review and studies have been selected and discussed based on preference/choices from the authors. We included studies that reported dysregulation of ncRNAs between BD patients and normal controls. We also included data regarding the expression profile of targets of ncRNAs whenever this data was provided in the original articles.

## 2. CircRNAs and BD

CircRNAs have a circular secondary structure. This structure is formed by the rear-splicing of a single-stranded linear RNA and the creation of a covalent link. These procedures lead to the formation of an encircled non-polyadenylated RNA structure. Compared with linear RNAs, circRNAs have higher stability resulting from a lack of accessible ends for exoribonucleases. They are about 0.1–10% of linear transcripts in eukaryotes [9]. CircRNAs participate in brain development and integrity of neurons [10,11]. A number of studies have demonstrated the release of brain-specific circRNAs into the peripheral blood in the course of neurological disease, potentiating these transcripts as biomarkers for showing disease progression or response to therapies [9]. Few studies have assessed the expression of circRNAs in brain samples of patients with BD obtained through autopsy. Luykx et al. have shown up-regulation of two circRNAs in brain tissues of bipolar patients. These transcripts were originated from the NEBL and EPHA3 loci, respectively [6]. The latter locus is involved in the development of CNS. Eph receptors of the protein-tyrosine kinase family participate in the production of neurotransmitters, construction of dendritic spines, and synaptic and postsynaptic events [12]. Moreover, they contribute to memory-associated functions [13] and modulation of anxiety [14], two functions that are disturbed in bipolar patients. Authors have suggested circRNA molecules as possible markers for the diagnostic assessment of patients with BD [6]. Zimmerman et al. have recently shown expression of circHomer1a, a neuron-associated circRNA in the frontal cortex. Notably, expression of this circRNA was decreased in both the prefrontal cortex (PFC) and induced pluripotent stem cell-originated neurons of patients with BD. CircHomer1a has been shown to regulate the expression of several splicing variants of genes participating in synaptic plasticity and psychiatric disorders. Thus, circHomer1a modulates synaptic gene activity and intellectual flexibility [15]. Table 1 and Table 2 summarize the results of studies that assessed the expression of circRNAs in BD.

## 3. LncRNAs and BD

Dysregulation of numerous lncRNAs has been described in peripheral blood and brain tissues of patients with BD. Hu et al. have profiled transcriptome in post-mortem brain tissues of patients with schizophrenia and BDs as well as healthy subjects. They reported differential expression of several long intergenic RNAs in various brain regions of bipolar patients. They showed that these lncRNAs have brain region-specific signatures and are mostly enriched in some pathways including immune system development and oligodendrocyte differentiation. Altered expression of these lncRNAs in patients was explained by modification of DNA methylation alteration [16]. Ji et al. have reported up-regulation of the XIST gene, the principal regulator of X chromosome inactivation (XCI) in the lymphoblastoid cells and brain tissues of female subjects with either BD or major depressive disorder. This up-regulation was accompanied by over-expression of the XCI escapee gene KDM5C. Authors have suggested that up-regulation of XIST might cause or result from delicate changes in XCI [17]. Table 3 summarizes the studies which reported up-regulation of lncRNAs in bipolar patients.

Ghafelehbashi et al. have assessed the expression of IFNG-AS1 lncRNA, and IFNG and IL-1B mRNAs in peripheral blood of BD patients compared with healthy subjects. They reported down-regulation of IFNG-AS1 in patients and its correlation with IFNG expression. Moreover, expression of IL-1B was decreased in patients compared with controls. Thus, inflammatory lncRNAs might participate in the pathogenesis of BD [21]. Hu et al. have reported down-regulation of ENSG00000228794 in patients with BD. This lncRNA resides in a genomic region that is linked with BD. ENSG00000228794 is possibly implicated in calcium ion transport, thus it can modulate synaptic plasticity [16]. Table 4 summarizes the results of studies that reported down-regulation of lncRNAs in BD.

## 4. miRNAs and BD

The expression profile of miRNAs has been vastly assessed in different biological sources of patients with BD including whole blood, lymphoblastoid cell lines, brain tissues or extracellular vesicles. Squassina et al. have assessed miRNAs signature in lymphoblastoid cell lines from bipolar patients who deceased by suicide and those with low risk of suicide. They reported higher miR-4286 levels while lower miR-186-5p in lymphoblastoid cell lines obtained from suicide attempters compared with the low-risk group and healthy controls. Conversely, expression of miR-4286 was reduced in postmortem brains of bipolar patients who attempted suicide compared with controls, yet it could not yield the level of significance. Exposure of human neural progenitor cells with lithium down-regulates expression of miR-4286 [23]. Lee et al. have reported abnormal expression of a number of miRNAs in the serum samples of bipolar patients. Among dysregulated miRNAs has been miR-7-5p which was up-regulated in bipolar patients [8]. Notably, miR-7 has been previously shown to suppress the healing of damaged peripheral nerves by altering the migration and proliferation of neural stem cells [8]. Moreover, this miRNA has been over-expressed in the neocortex of superior temporal lobes of patients with Alzheimer’s disease [24]. Choi et al. have extracted extracellular vesicles (EVs) from the anterior cingulate cortex. They reported over-expression of miR-149 in bipolar patients compared to controls. They also validated dysregulation of both miRNAs in EVs extracted from brains of an animal model of depressive-like manners [25]. Figure 1 shows the molecular mechanisms of participation of miR-34a in the pathogenesis of BD.

Table 5 summarizes the studies which reported up-regulation of miRNAs in BD.

Pisanu et al. have assessed miRNA profile in lymphoblastoid cell lines from BD patients who responded to lithium versus non-responders. They described differential expression of 31 miRNAs between these groups, among them were miR-320a and miR-155-3p. Expression of hsa-miR-320a was significantly lower in responders. Notably, targets of this miRNA participate in neuronal survival and differentiation, apoptosis, and plasticity of synapses [29]. Zhang et al. have demonstrated deceased circulating levels of miR-134 in bipolar patients as well as patients with schizophrenia or major depressive disorder compared with normal controls. Yet, the most significant downregulation of this miRNA has been described in major depressive disorder [45]. Table 6 summarizes the list of down-regulated miRNAs in BD.

Lim et al. have appraised the expression of miRNAs in peripheral blood of bipolar manic patients after 12 weeks of receiving asenapine or risperidone. They reported differential expression of several miRNAs [52]. Table 7 summarizes these miRNAs.

## 5. Discussion

Several studies have reported aberrant expression of ncRNAs in bipolar patients. Moreover, the expression of ncRNAs is influenced by drugs used for these patients. For instance, a combination of drugs including lithium, valproate, lamotrigine and quetiapine has been shown to alter the expression of several genes including miRNAs in cultured human neurons. Among the differentially expressed genes have been miR-128 and miR-378 whose targets are enriched in neuron projection development and axonogenesis [53]. Thus, ncRNAs not only are involved in the pathogenesis of BD but also they might participate in the determination of response to prescribed drugs. NcRNA profiling has revealed specific alterations in certain lncRNAs and miRNAs in the manic state indicating a possible role of these transcripts in the determination of disease status [18]. Notably, lncRNAs have been the largest group of differentially expressed ncRNAs [18]. Such state-specific transcript signature potentiates ncRNAs as preferable biomarkers for early diagnosis of BD.

Next generation sequencing technique has facilitated the identification of putative biomarkers for discrimination of bipolar patients from healthy subjects. A representative of this kind of experiment is the study conducted by Lee et al. which identified over-expression of miR-7-5p, miR-23b-3p, miR-142-3p, miR-221-5p, and miR-370-3p in bipolar patients compared with healthy individuals. The diagnostic accuracy of this panel of miRNAs was estimated to be 0.907 [8].

Dysregulated ncRNAs in bipolar patients have been enriched in several neuron-related pathways such as GABAergic and glutamatergic synapses, morphine addiction pathway, redox modulation as well as TGF-β, Wnt, Akt/PI3K, Hippo and FoxO pathways. Significance of a number of these pathways such as GABAergic and glutamatergic synapses signaling and TGF-β, Hippo and FoxO pathways have been recognized in the pathogenesis of BD [54,55]. The relevance of other pathways with this disorder should be appraised in future studies. Another functional annotation analysis of the differentially expressed coding and non-coding genes between patients with BD and healthy controls has shown remarkable enrichments of cellular pathways associated with angiogenesis and vascular system evolution [6]. The largest GWAS conducted in BD has reported that the most significant loci have been related to ion channels, neurotransmitter transporters and synaptic components. Yet, this study has not reported any indication for involvement of angiogenesis or vascular related loci in BD [56]. Pathway analysis revealed nine significantly enriched gene sets, including regulation of insulin secretion, circadian rhythm, and endocannabinoid signaling [56]. Notably, insulin resistance signaling pathway and circadian rhythm have been among the related pathways with dysregulated miRNAs in BD [8,23]. Finally, top genes existing in these pathways have been shown to encode Ca2+ and K+ channel subunits, MAPK and GABA-A receptor subunits [56], the latter being recognized as one of the most important pathways enriched among dysregulated ncRNAs in BD [8].

However, different studies have indicated abnormal activity of various signaling pathways in BD including immune response pathways [57], neuroplasticity, circadian rhythms and GTPase binding [58] and G protein-receptor dysregulation [59]. Such a heterogeneous range of biological pathways involved in BD might be related to distinct brain areas assessed in these investigations. Imminent investigations integrating particularly large sample sizes of patients with BD and comparison of transcriptome of coding and ncRNAs in different parts of the brain are required to find the most relevant pathways. 

Dysregulation of ncRNAs has been reported in other brain disorders as well. For instance, assessment of lncRNA signature using high-throughput sequencing has led to the recognition of aberrantly expressed lncRNAs in acute ischemic stroke. ENSG00000226482 has been among up-regulated lncRNAs. This lncRNA has a potential role in activation of the adipocytokine signaling [60]. Moreover, another experiment in the animal model of blast traumatic brain injury has shown elevation of plasma amounts of a brain-enriched miRNA, namely miR-127. This study has concluded that levels of sphingolipids, miR-128, and the let-7 family can show the presence of could blast traumatic brain injury. Moreover, a number of other miRNAs have been shown to serve as markers for a global level of damage after blast injury [61]. Moreover, miRNAs have been shown to serve as diagnostic markers for cognitive impairment. Certain panels of miRNAs have high sensitivity and specificity values in this regard [62].

Notably, several dysregulated ncRNAs in BD, are also dysregulated in other neuropsychiatric conditions such as schizophrenia or Alzheimer’s disease. Although this observation supports their potential roles in synaptic plasticity or neurodevelopment, it complicates the design of disease-specific diagnostic panels for BD.

Among dysregulated miRNAs in peripheral blood of patients with BD have been miR-128, miR-133b, miR-29a, miR-370, miR-451, miR-874 and miR-9* which have been recognized as brain-enriched miRNAs [63]. 

Taken together, circRNAs, lncRNAs and miRNAs are regarded as potential contributors in the pathology of BD and putative biomarkers for diagnosis of this disorder. Their participation in the response of patients to the prescribed medications and their potential as therapeutic targets have been less studied. Thus, these research areas should be explored in future studies.

NcRNAs are superior to transcripts of standard genes in the field of biomarker study as they represent the final step of function of the gene. As transcripts of standard genes should be translated to proteins to exert their function, the transcript level might not reflect the final level of the functional molecule. Moreover, miRNAs represent important regulators of gene expression as they can target several transcripts. 

Finally, studies reporting dysregulation of ncRNAs in BD have some limitations. For instance, they often suffer from various confounding factors. This is especially true for postmortem brain studies. Moreover, most of the studies reviewed in this manuscript may not have sufficient statistical power due to their small sample sizes. Analysis and interpretation of differences between data of postmortem brains and blood samples, differences between expression data on ncRNA and protein coding genes, and matching with GWAS-identified loci are other research fields that should be explored in future studies. 

## 6. Conclusions

NcRNAs are potential markers for neurological disorders such as BD. Several ncRNAs have been found to be dysregulated in blood samples of bipolar patients. Molecular studies for identification of the mechanism of dysregulation of these transcripts in bipolar patients would facilitate the development of new therapeutic strategies.

## Figures and Tables

**Figure 1 ijms-22-05156-f001:**
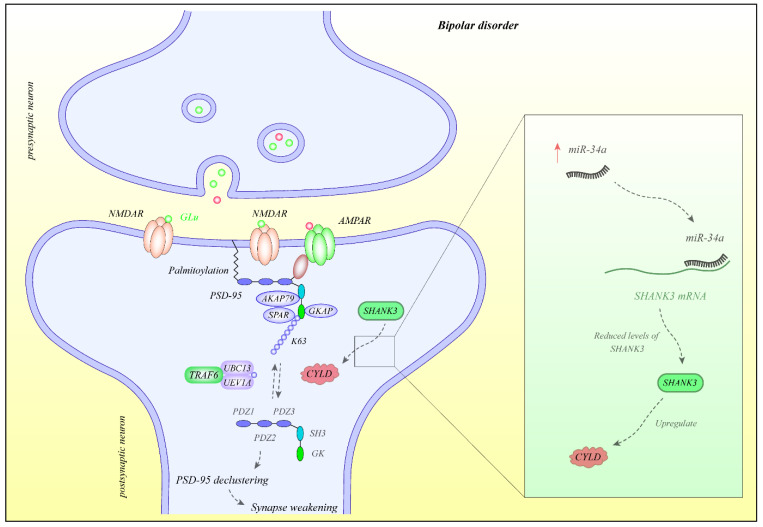
Expression of miR-34a is increased in BD. This miRNA binds with 3’ UTR of SHAK3 to decrease its expression. Expression of this protein is correlated with CYLD levels [26,27]. CYLD is a deubiquitinase that targets PSD-95. The latter protein participates in the maturation and function of synapses [28].

**Table 1 ijms-22-05156-t001:** Summary of function of up-regulated circRNAs in BD.

circRNA	Number of Clinical Samples	Type of Study	False Discovery Rate	Function	Ref
cNEBL	postmortem human medial frontal gyrus tissues from BPD cases (*n* = 4) and normal subjects (*n* = 4)	High throughput analysis	FDR < 0.05	-	[6]
cEPHA3		High throughput analysis	FDR < 0.1	EPHA3 participates in the neurodevelopment.	

**Table 2 ijms-22-05156-t002:** Summary of function of down-regulated circRNAs in BD (BD: bipolar disorder, SZ: schizophrenia).

circRNA	Number of Clinical Samples	Targets/Regulators	*p* Value	Function	Ref
circHomer1a	Human OFC postmortem brain tissues of BD patients (*n* = 32) and healthy controls (*n* = 34)	RNA-binding protein HuD	*p* < 0.01	circHomer1a is originated from HOMER1, a gene regulating neuronal excitability and synaptic plasticity. This gene is down-regulated in the OFC and stem cells-originated neurons of BD patients.	[15]

**Table 3 ijms-22-05156-t003:** Summary of function of up-regulated lncRNAs in BD (BD: bipolar disorder, SZ: schizophrenia).

lncRNA	Number of Clinical Samples	Assessed Cell Line	Targets/Regulators	*p* Value	Type of Study	Function	Ref
PTCSC3	Whole blood sample of BD patients with manic episode (*n* =13)	-	-	*p* = 2.39 × 10^−4^	High throughput analysis	PTCSC3 gene is reported to be associated with thyroid cancer.	[18]
CCAT2	Peripheral blood specimens BD patients (*n* = 50) and healthy subjects (*n* = 50)	-	-	*p* = 0.006	Candidate molecule analysis	CCAT2 is an oncogenic lncRNA in numerous neoplasms that enhances cell proliferation and suppresses apoptosis.	[19]
TUG1		-	*-*	*p* < 0.001	Candidate molecule analysis	TUG2 is an oncogenic lncRNA in numerous neoplasms that enhances cell proliferation and suppresses apoptosis.	[19]
PANDA		-	*-*	*p* = 0.004	Candidate molecule analysis	PANDA is an oncogenic lncRNA in numerous neoplasms that enhances cell proliferation and suppresses apoptosis.	[19]
DISC2	Peripheral blood mononuclear cells (PBMCs) of BD patients (*n* = 50) and controls (*n* = 50)	-	hsa-miR-92a-2-5p, hsa-miR363-5p, hsa-miR-1285-3p and hsa-miR-1268a	*p* = 0.0015	Candidate molecule analysis	DISC2 may regulate DISC1 expression.	[20]
XIST	Lymphoblastoid cells were from healthy females (*n* = 36) and female patients with either BD or recurrent major depression (*n* = 60)	Lymphoblastoid cell lines	TSIX, FTX, JPX	*p* = 1 × 10^−7^	Candidate molecule analysis	XIST is the master gene for XCI.	[17]
FTX		Lymphoblastoid cell lines	XIST	*p* < 0.1	Candidate molecule analysis	FTX is a positive regulator of XIST expression.	[17]

**Table 4 ijms-22-05156-t004:** Summary of function of down-regulated lncRNAs in BD (BD: bipolar disorder, SZ: schizophrenia).

lncRNA	Number of Clinical Samples	Assessed Cell Line	Targets/Regulators	Signaling Pathways	*p* Value	Type of Study	Function	Ref
OIP5-AS1	Peripheral blood samples of BD patients (*n* = 50) and healthy controls (*n* = 50)	-	-	-	*p* = 0.001	Candidate genes analysis	OIP5-AS1 is as an oncogene that enhances cell proliferation and suppresses of apoptosis.	[19]
IFNG-AS1	Blood samples of BD patients (*n* = 30) and healthy control individuals (*n* = 32)	-	IFNG	-	*p* < 0.0001	High throughput analysis	IFNG-AS1 facilitates IFN-γ expression through association with the WDR methyltransferases and subsequent increase in H3K4 methylation at the IFNG locus.	[21]
ENSG00000228794	Post-mortem brain samples of patients with SZ and BD and control subjects (*n* = 82)	-	-	Calcium signaling	*p* < 0.05	High throughput analysis	ENSG00000228794 is located in a genomic region linked with BD and partakes in calcium ion transport.	[16]
TSIX	Lymphoblastoid cells were from healthy females (*n* = 36) and female patients with either BD or recurrent major depression (*n* = 60)	Lymphoblastoid cell lines	XIST	-	*p* < 0.01	Candidate genes analysis	TSIX is a negative regulator of XIST expression.	[17]
MALAT1	Peripheral blood samples of BD patients (*n* = 50) and healthy controls (*n* = 50)	PBMCs	hsa-miR-17-5p, hsa-miR-106a-5p, hsa-miR-30c-5p, hsa-miR-20b-5p, hsa-miR-503, hsa-miR-92b-3p, hsa-miR-1224-3p	-	*p* < 0.0001	Candidate molecule analysis	MALAT1 takes part in the regulation of genes involved in synaptogenesis.	[22]

**Table 5 ijms-22-05156-t005:** Summary of function of up-regulated miRNAs in BD (BD: bipolar disorder, SZ: schizophrenia).

microRNA	Number of Clinical Samples	Assessed Cell Line	Targets/Regulators	Signaling Pathways	*p* Value	Type of Study	Function	Ref
miR-7-5p	Whole blood samples of BD-II patients (*n* = 102) and controls (*n* = 118)	-	BDNF	GABAergic and glutamatergic synapses and TGF-beta, Hippo, and FoxO signaling	*p* < 0.001	High throughput analysis	miR-7 has a role in inhibition of the repair of peripheral nerve damage by affecting the migration and proliferation of neural stem cells.	[8]
miR-142-3p		-	TNF-α		*p* < 0.0001	High throughput analysis	miR-142-3p may modulate the BMAL1 gene and regulate circadian functions.	[8]
miR-221-5p		-	-		*p* < 0.0001	High throughput analysis	miR-221 is potentially involved in atherosclerosis.	[8]
miR-370-3p		-	-		*p* < 0.0001	High throughput analysis	miR-370 is reduced in brain tissue of depressed animals.	[8]
miR-23b-3p		-	-		*p* = 0.006	High throughput analysis	miR-23b may have an anti-inflammatory role in central nervous system inflammation.	[8]
miR-4286	Lymphoblastoid cell line cultures from patients with BD who died by suicide (SC, *n* = 7) and with low risk of suicide (LR, *n* = 11) and 12, non-suicidal controls	Lymphoblastoid cell lines (LCLs)	PRKAB2, PTPRF, PIK3R3, CREB1, PPARGC1B, PIK3R1, CREB3L2, PTPA, PTEN RELA, PRKAG1, PTPN11, PRKCB, SOCS3, INSR, PYGB, PPARA	Insulin resistance signaling pathway	*p* = 0.000043	High throughput analysis	miR-4286 might be a specific biomarker of suicide.	[23]
hsa-miR-155-3p	Lymphoblastoid cell line cultures from BD patients excellent responders (ER, *n* = 12) and non-responders (NR, *n* = 12) to lithium	Lymphoblastoid cell lines	SP4	-	*p* = 0.0003	High throughput analysis	hsa-miR-155-3p was up-regulated in ER. It partakes in inflammatory response and modulates differentiation and activation of innate and adaptive immune systems.	[29]
miR-223	Orbitofrontal cortex of SZ (*n* = 29; 20 males and 9 females), BD (*n* = 26; 12 males and 14 females), and unaffected controls (*n* = 25; 21 males and 4 females)	-	GRIN2B, GRIA2	-	*p* < 0.001	High throughput analysis	miR-223 regulates glutamate receptors. miR-223 expression is negatively correlated with levels of its targets GRIN2B and GRIA2.	[30]
miR-193b-3p		-	-	-	*p* < 0.05	High throughput analysis	miR-193a-3p was upregulated in both BD and SZ.	[30]
miR-330-3p		-	-	-	*p* < 0.05	High throughput analysis	miR-330-3p has been over-expressed in the blood of subjects with BD and monopolar depression.	[30]
miR-28a-3p		-	-	-	0.05 < *p* <0.10	High throughput analysis	miR-28a-3p is in the same family as miR-708, a miRNA that is associated with risk of BD.	[30]
miR-1260		-	-	-	*p* < 0.05	High throughput analysis	-	[30]
miR-185-5p	Plasma samples of patients with BD type I (*n* = 69; 15 depressed, 27 manic, 27 euthymic) and healthy controls (*n* = 41)	-	Tyrosine kinase receptor type 2	PI3K-Akt	*p* = 0.001	High throughput analysis	miR-185-5p is a target miRNA for depression.	[31]
miR-29a-3p	Peripheral blood of BD I patients (*n* = 58, 19 manic, 39 euthymic) and healthy controls (*n* = 51)	-	-	PI3K-Akt, TGF-beta	*p* = 0.035	Candidate analysis	-	[32]
miR125a-3p					*p* = 0.014			
miR-106b-5p		-	IL-10	PI3K-Akt, TGF-beta	*p* = 0.014	Candidate analysis	miR-106 might be involved in immunomodulatory aspects of BD.	[32]
miR-107		-	GRIN2A, SLC1A4	PI3K-Akt, TGF-beta	*p* = 0.011	Candidate analysis	miR-107 is up-regulated in manic and euthymic patients.	[32]
hsa-miR-150-5p, hsa-miR-25-3p, hsa-miR-451a, hsa-miR-144-3p	Plasma samples from drug-free psychotic bipolar patients (*n* = 15) and HC (*n* = 9)	-	-	-	*p* < 0.01	High throughput analysis	These miRNAs were upregulated in patients.	[33]
hsa-miR-4516, hsa-miR-6808-5p, hsa-miR-7977, hsa-miR-1185-2-3p, hsa-miR-6791-5p, hsa-miR-3194-5p, hsa-miR-6090, hsa-miR-3135b	Peripheral blood EV of BD patients (*n* = 20) and age- and sex-matched normal controls (*n* = 21)	-	-	Axon guidance mediated by netrin, endothelin signaling pathway, 5HT2 type receptor-mediated signaling, beta1 and beta2 adrenergic receptor and the androgen receptor signaling pathways	*p* < 0.05	High throughput data mining	These miRNAs are related to neuron development.	[34]
hsa-miR-29c-3p			-	-	*p* = 0.010514	High throughput data mining	Increased levels of miR-29c have been detected in EVs isolated from post-mortem prefrontal cortex (BA9) of patients.	[34]
hsa-miR-7975			-	-	*p* = 0.048192	High throughput data mining	hsa-miR-7975 is associated with the brain.	[34]
hsa-miR-21-5p			NTN1, NTNG1,	-	*p* = 0.028475	High throughput data mining	The increased levels of miR-22 in EVs are supported by findings of upregulation of these miRNAs in the prefrontal cortex of patients with BD.	[34]
hsa-miR-142-3p			NTN3, NTNG1, NTNG2	-	*p* = 0.019266	High throughput data mining	NTN3, NTNG1, NTNG2 are targeted by hsa-miR-142-3p.	[34]
hsa-miR-22-3p, hsa-miR-92a-3p			NTN3, NTN4, NTNG1, NTNG2	-	*p* < 0.05	High throughput data mining	NTN3, NTN4, NTNG1, NTNG2 are targeted by hsa-miR-22-3p, hsa-miR-92a-3p.	[34]
hsa-miR-198	3 frontal cortex miRNA expression datasets (N for BD = 30–36 per dataset, N for controls = 28–34 per dataset)	-	GPX4	Redox modulation pathways	*p* < 0.05	High throughput data mining	These are the top 10th percentile of up-regulated miRNAs that target redox modulators ranked for their ability to discriminate between BD and controls in Vladimirov dataset.	[35]
hsa-miR-601		-	ATP2B4		*p* < 0.05	High throughput data mining		[35]
hsa-miR-659		-	SOD2, ATP5A1		*p* < 0.05	High throughput data mining		[35]
hsa-miR-192		-	TXN2, UQCRC2, ATP5L, PXDN, TXNIP, COX5A, TXN2		*p* < 0.05	High throughput data mining		[35]
hsa-miR-346		-	NDUFA1, COX5A		*p* < 0.05	High throughput data mining		[35]
hsa-miR-9*		-	ATP5F1		*p* < 0.05	High throughput data mining		[35]
hsa-miR-301a		-	ATP5B, COX10, TXNIP, BCL2L11, NDFUA7, TXNRD3, COX7A2, OXA1L, MGST1, PXDN, SOD2, COX5B, NDUFA5, UQCRQ		*p* < 0.05	High throughput data mining		[35]
hsa-miR-199a-3p		-	ATP5B, UQCRC2, COX10, TXNIP, BCL2L11, PTGS2, GLRX2, NDUFA2, NDUFC2, NDUFA12		*p* < 0.05	High throughput data mining		[35]
hsa-miR-34a		-	TXNIP, NDUFS1, SOD2, PRDX5, NDUFV1		*p* < 0.05	High throughput data mining		[35]
hsa-miR-145		-	NDUFS1, NDUFA4		*p* < 0.05	High throughput data mining		[35]
hsa-miR-27a		-	PPA1, TXNIP, ATP2B4, ATP5SL, PRDX1, PRDX4, FOXO3, NDUFA8, CAT, PXDN, SOD1, FOXO1, NDUC2, GSTO1, ATP5G3, NDUFV3, NDUFS2, NDUFS4, NDUFV1, PRDX3, PPA1, GLRX5		*p* < 0.05	High throughput data mining		[35]
hsa-miR-92a-1*		-	BCS1L, NDUFS1, SDHB, OXA1L		*p* < 0.05	High throughput data mining		[35]
hsa-miR-103		-	MGST1, TXN2, NDUFS8, ATP5B, PRDX4, ATP5A1, OXA1L, NDUFS2, NOS3, COX5A, TXNRD3		*p* < 0.05	High throughput data mining		[35]
hsa-miR-196b		-	ATP5G3, MT-ATP6, BCL2L2, BCL2L12, ATP2B4, GLRX3, NDUFC2, OXA1L, NDUFV3		*p* < 0.05	High throughput data mining		[35]
hsa-miR-449a		-	ATP5H, TXNIP, FOXO1, BCL2L11, ATP6V0A2, NDUFS1		*p* < 0.05	High throughput data mining		[35]
hsa-miR-196a		-	GPX1, ATP5G3, UQCRC2, TXR1, MTATP8, MT-ATP6, BCL2L12, ATP2B4, GLRX3, FOXO1, OXA1L, NDUFV3, NDUFC2, GSTK1		*p* < 0.05	High throughput data mining		[35]
hsa-miR-675		-	SOD2, PXDN		*p* < 0.05	High throughput data mining		[35]
hsa-miR-184		-	BCL2A1		*p* < 0.05	High throughput data mining		[35]
hsa-miR-200c		-	MT-ATP6, MGST1		*p* < 0.05	High throughput data mining		[35]
hsa-miR-200b		-	BCL2, GLRX5		*p* < 0.05	High throughput data mining		[35]
miR-149	Extracellular vesicles (EVs) extracted from Human Postmortem Anterior Cingulate Cortices (BA24) Diagnosed With BD (*n* = 4) and Non-Psychiatric Control Cases (*n* = 6)	Neuronal and glial cells	-	AKT1	*p* = 0.0046	Candidate analysis	miR-149 suppresses glial proliferation.	[25]
miR-30d-5p	Blood samples of MD patienst (*n* = 20) and BD patienst (*n* =20, 10 type I and 10 type II) and healthy controls (*n* = 20, 15 females, 5 males)	-	-	-	*p* = 0.028	High throughput analysis	The blood expression of miR-30d-5p was increased also in MD patients after AD treatment.	[36]
miR140-3p		-	-	-	*p* = 0.027	High throughput analysis	The blood expression of miR140-3p was increased also in MD patients after AD treatment.	[36]
miR-330-5p		-	HTR2C, MAOA, DRD1, CAMKK2, NTRK3, CLOCK, CREB1, GABRA2, CNR1, MTHFR	-	*p* = 0.030	High throughput analysis	miR-330-5p regulates many targets participating in neuronal plasticity and neurodevelopment.	[36]
miR-21-3p		-	-	-	*p* = 0.043	High throughput analysis	miR-21-3p is decreased in MD fibroblast cultures.	[36]
miR-378a-5p		-	-	-	*p* = 0.042	High throughput analysis	miR-378a-5p is mainly involved in lipid and metabolism homeostasis.	[36]
hsa-miR-345-5p		-	HTR2C, MAOA, DRD1, CAMKK2, NTRK3, CLOCK, CREB1, GABRA2, CNR1, MTHFR	-	*p* = 0.010	High throughput analysis	miR-345- 5p is predicted to regulate several target genes with a putative role in the shared pathogenetic mechanisms between MD and BD.	[36]
miR-15b	Blood of unaffected individuals at higher genetic risk of developing a mood disorder (*n* = 34) and control subjects (*n* = 46)	-	-	PI3K/Akt, PTEN	*p* = 0.0166	Candidate gene analysis (20 miRNAs)	miR-15b was over-expressed in the high-risk persons. It is involved in metabolism, angiogenesis, stress response, cancer, cardiovascular disease and neurodegenerative conditions.	[37]
miR-132		-	-	PI3K/Akt	*p* = 0.0249	Candidate gene analysis (20 miRNAs)	miR-132 was over-expressed in the high-risk persons. miR-132 is transcribed from a cluster of miRNAs that partake in neuronal development and function.	[37]
miR-652		-	GABARB2, GABARB3, 5-HT1D, DISC1	-	*p* = 0.01076	Candidate gene analysis (20 miRNAs)	miR-652 was up-regulated in the high-risk individuals. miR-652 plays a central role in myeloid development.	[37]
miR-34a	Postmortem human brain samples from the cerebellum (lateral cerebellar hemisphere; 34 control and 29 BD samples)	-	ANK3, CACNB3, DDN, SHANK3	WNT, cadherin	*p* < 0.01	Candidate analysis	miR-34a expression is inversely associated with expression of ANK3 and CACNB3.	[27,38]
miR-17-5p	Human prefrontal cortex (Brodmann area 10) of 15 SZ, 15 MDD, 15 BD, and 15 controls	-	-	-	*p* = 0.0028	High throughput analysis	-	[39]
miR29c-3p					*p* = 0.049			
miR-106b-5p					*p* = 0.021			
miR-579					*p* = 0.0092			
miR-29c	Postmortem Human Prefrontal Cortex (Brodmann area 9, BA9) 8 SZ, 9 BD, and 13 controls	-	-	Wnt	*p* = 0.0237	High throughput analysis	miR29c is induced by canonical Wnt signaling.	[40]
hsa-miR-188-5p, hsa-miR-196b, hsa-miR-32*, hsa-miR-187, hsa-miR-383, hsa-miR-297, hsa-miR-876-3p, hsa-miR-490-5p, hsa-miR-449b, hsa-miR-513-5p	Dorsolateral prefrontal cortex tissue of control (*n* = 34), bipolar (*n* = 31), and schizophrenic (SZ, *n* = 35) subjects	-	-	-	*p* < 0.05	High throughput analysis	-	[41]
hsa-miR-504	Postmortem DLPFC sections from 35 cases with schizophrenia 35 cases with BD	-	-	-	*p* = 0.00003	High throughput analysis	-	[42]
hsa-miR-145*					*p* = 0.00080			
hsa-miR-22*					*p* = 0.00106			
hsa-miR-145					*p* = 0.00177			
hsa-miR-133b					*p* = 0.00190			
hsa-miR-154*					*p* = 0.00195			
hsa-miR-889					*p* = 0.00321			
miR-34a	20 LCLs derived from bipolar I disorder (BPI) family members with and without LiCl treatment in culture	Lymphoblastoid cell lines (LCLs)	AP2A1, AP2S1, CD2AP, EIF1, and VCL	-	*p* = 0.023917	Candidate analysis (13 miRNAs)	miR-34a, miR-152, miR-155, and miR-221 were consistently up-regulated at treatment time point day 4 and day 16.	[43]
miR-152					*p* = 0.000405			
miR-155					*p* = 0.012045			
miR-221					*p* = 0.000073			
miR-195-5p,	Skin biopsies of 3 control and 3 BP patient	Pluripotent Stem Cell-derived neurons	AXIN2, BDNF, CACNA1E, MIB1, NLGN1 and RELN	Axon guidance, Mapk, Ras, Hippo, Neurotrophin and Wnt signaling pathway	*p* < 0.05	Candidate molecule analysis(58 miRNAs)	-	[44]
miR-382-5p			SYT4					
miR-128-3p, miR-138-2-3p, miR-487b-3p, miR-744-3p			-					

**Table 6 ijms-22-05156-t006:** Summary of function of down-regulated miRNAs in BD (BP: bipolar disorder, SZ: schizophrenia).

microRNA	Number of Clinical Samples	Assessed Cell Line	Targets/Regulators	Signaling Pathways	*p* Value	Type of Study	Function	Ref
miR-320a	BD patients (excellent responders, *n* = 12; non-responders, *n* = 12) to lithium	Lymphoblastoid cell lines	CAPNS1	-	*p* < 0.0001	High throughput analysis	Participates in response to lithium	[29]
miR-134	Whole blood samples of BD (*n* = 50) and controls (*n* = 50)	-	cAMP response element-binding protein (CREB)	-	*p* = 2.25 × 10^−5^	Candidate molecule analysis	miR-134 regulates dendritic spine development and plasticity.	[45]
miR-186–5p	LCLs from patients with BD who deceased by suicide (SC, *n* = 7) and with low risk of suicide (LR, *n* = 11) and 12, non-suicidal controls	Lymphoblastoid cell lines (LCLs)	-	-	*p* = 0.032	High throughput analysis	miR-186–5p was lower in lithium-treated LCLs from SC compared to controls.	[23]
miR-484	Plasma samples of patients with BD type I and healthy controls (*n* = 41)	-	-	PI3K-Akt	*p* < 0.001	High throughput analysis	miR-484 is linked with neurogenesis, mitochondrial network and redox modulations	[31]
miR-142-3p		-	-	PI3K-Akt	*p* = 0.001	High throughput analysis	miR-142-3p regulates signaling pathways during embryonic development and homeostasis.	[31]
miR-652-3p		-	-	PI3K-Akt	*p* < 0.001	High throughput analysis	miR-652 is linked with immune system and oxidative stress.	[31]
hsa-miR-363-3p, hsa-miR-4454 + has-miR-7975, hsa-miR-873-3p, hsa-miR-548al, hsa-miR-598-3p, hsa-miR-4443, hsa-miR-551a, hsa-miR-6721-5p	Plasma samples from drug-free psychotic BD cases (*n* = 15) and HC (*n* = 9)	-	-	-	*p* < 0.01	High throughput analysis	These miRNAs were downregulated in patients.	[33]
hsa-miR-1281, hsa-miR-6068, hsa-miR-8060, hsa-miR-4433a-5p, hsa-miR-1268b, hsa-miR-1238-3p, hsa-miR-188-5p, hsa-miR-6775-5p, hsa-miR-6800-3p, hsa-miR-3620-5p, hsa-miR-451a, hsa-miR-1227-5p, hsa-miR-7108-5p, hsa-miR-671-5p, hsa-miR-6727-5p, hsa-miR-6125, hsa-miR-6821-5p	Peripheral blood EVs from BD patients (*n* = 20) and age- and sex matched normal subjects (*n* = 21)	-	-	Axon guidance mediated by netrin, endothelin signaling, 5HT2 type receptor-mediated signaling, beta1 and beta2 adrenergic receptor pathways, and the androgen receptor signaling pathway	*p* < 0.05	High throughput analysis	These miRNAs were nominally downregulated between patients and controls. Pathway analyses identified some brain-relevant mechanisms enriched in these miRNAs, including axon guidance by netrin and the serotonin receptor pathway.	[34]
hsa-miR-5739		-	-	-	*p* = 0.024667	High throughput analysis	miR-5739 is suggested to be highly associated with the brain.	[34]
hsa-miR-133a-3p	Peripheral blood EVs from BD type I (*n* = 20) and age- and sex matched healthy controls (*n* = 21)	-	NTN1, NTN3, NTNG1, NTNG2	-	*p* < 0.05	High throughput data mining	-	[35]
hsa-miR-299-5p	3 frontal cortex miRNA expression datasets	-	SOD2, GPX4	Redox modulation pathways	*p* < 0.05	High throughput data mining	These are the top 10th percentile of decreased miRNAs that target redox modulators ranked for their ability to discriminate between BD and controls in Miller dataset.	[35]
hsa-miR-197		-	SOD1, GCLC, TXN, COX8A, ATP2B4		*p* < 0.05	High throughput data mining		[35]
hsa-miR-23a		-	NDUFA2, PPA1, GCLM, PTGS1, SOD2, PRDX4, PXDN, TTN, UQCRQ, NDUFV1, PRDX3, NDUFA3, TXNIP, ATP5O, TXNRD1		*p* < 0.05	High throughput data mining		[35]
hsa-miR-450a		-	GCLC, NDUFA10,ATP5C1		*p* < 0.05	High throughput data mining		[35]
hsa-miR-17		-	ATP5B, TXN, NDUFA10, TXNIP, MTATP6, BCL2L11, NDUFS1, OXA1L, ATP2B4, BCl2L13, TXN2, SOD2, SDHB, PXDN, FOXO1, BCL2, UQCRFS1, RXNRD2, GPX2, TXNRD2		*p* < 0.05	High throughput data mining		[35]
hsa-miR-944		-	FOXO1		*p* < 0.05	High throughput data mining		[35]
hsa-miR-19b		-	GCLC, ATP2B4, NDUFB2, COX6A1, FOXO3, PXDN, NDUFS3, COX10, NDUFB2		*p* < 0.05	High throughput data mining		[35]
hsa-miR-503		-	COX10, NDUFS1,PXDN		*p* < 0.05	High throughput data mining		[35]
hsa-miR-7		-	NDUFA4, SDHC, ATP5S, FOXO6, NDUFS1, GCLM, COX4I1, ATP2B4, TXN2, GSR, ATP5F1, SDHB, NDUFC2, PPA1, PRDX1		*p* < 0.05	High throughput data mining		[35]
hsa-miR-199a-5p		-	NDUFA13, MGST2		*p* < 0.05	High throughput data mining		[35]
hsa-miR-484		-	NOS3, PRDX1, COX7A2L, UQCRQ, GSTO1, UQCRFS1, ATP5J, BCL2L1, COX8A, PRDX1, MTATP6, PRDX4, COX5A, UQCRQ		*p* < 0.05	High throughput data mining		[35]
hsa-miR-424		-	NDUFS1, COX7A2L, BCL2L11, UQCRH		*p* < 0.05	High throughput data mining		[35]
miR-499	Peripheral blood of adult women only, 17 UP (age: 50 ± 17) and 15 BP (age: 33 ± 13) patients	-	-	-	*p* = 0.008	Candidate molecule analysis	miR-499 is down-regulated in depression episodes of the BD patients compared with remission phase.	[46]
miR-708		-	-	-	*p* = 0.02	Candidate molecule analysis	miR-708 is down-regulated in depression episodes of the BD patients compared with remission phase.	[46]
miR-1908		-	KLC2	-	*p* = 0.004	Candidate molecule analysis	miR-1908 is down-regulated in depression episodes of the BD patients compared with remission phase. It is involved in lipid metabolism. Overexpression of miR-1908 in multipotent adipose-derived stem cells suppressed adipogenic differentiation and increased cell proliferation.	[46]
miR-1908-5p	Two human NPC lines derived from dermal fibroblasts of either a control or a BD subject, treated with vehicle or 1 mM lithium or valproate for a week	Human neural progenitor cells (NPCs)	DLGAP4, GRIN1, STX1A, CLSTN1, GRM4	NF-kappaB	*p* < 0.05	Candidate molecule analysis	miR-1908 is an intronic miRNA of the fatty acid desaturase 1 (FADS1) gene.	[47]
miR-132	Human post-mortem anterior cingulate cortex (AnCg) tissue. (*n* = 8, BP; *n* = 15, MDD; *n* = 14, Control)	-	-	-	*p* < 0.05	Candidate molecule analysis (29 miRNAs)	-	[48]
miR-133a		-	-	-	*p* < 0.05	Candidate molecule analysis (29 miRNAs)	While miR-133b levels did not change, miR-133a was differentially expressed in the AnCg of cohort of BP patients.	[48]
miR-212		-	-	-	*p* < 0.05	Candidate molecule analysis (29 miRNAs)	miR-132 and miR-212 have been previously identified as differentially expressed in the DLPFC of SZ patients.	[48]
miR-34a		-	NCOA1, PDE4B	-	*p* < 0.05	Candidate molecule analysis (29 miRNAs)	miR-34a expression is dysregulated in SZ and BP patients. miR-34a has been linked to acute responses to stress.	[48]
miR-145-5p	Human prefrontal cortex (Brodmann area 10) of 15 SZ, 15 MDD, 15 BD, and 15 controls	-	-	-	*p* = 0.0069	High throughput	-	[39]
miR-485-5p					*p* = 0.036			
miR-370					*p* = 0.041			
miR-500a-5p					*p* = 0.041			
miR-34a-5p					*p* = 0.048			
hsa-miR-454*	Postmortem DLPFC tissues of individuals with schizophrenia (SZ, *n*= 35) and BD (*n* = 35)	-	-	-	*p* = 0.00004	High throughput	-	[42]
hsa-miR-29a					*p* = 0.00005			
hsa-miR-520c-3p					*p* = 0.00018			
hsa-miR-140-3p					*p* = 0.00053			
hsa-miR-767-5p					*p* = 0.00102			
hsa-miR-874					*p* = 0.00181			
hsa-miR-32					*p* = 0.00209			
hsa-miR-573					*p* = 0.00227			
miR-134	Plasma sample of drug-free bipolar I patients (14 men and 7 women) and controls (*n* = 21)	-	Limk1	-	*p* = 0.009	Candidate molecule analysis	miR-134 regulates dendritic spine development though Limk1, that controls synaptic development, maturation and/or plasticity.	[49]
miR-346	DLPFC samples of SZ patients (*n* = 35), BD (*n* = 32), normal subjects (*n* =34)	-	CSF2RA	-	*p* = 0.086	Candidate molecule analysis	miR-346 gene lies in intron 2 of the GRID1 gene, which has been proposed to be important in SZ susceptibility.	[50]
miR-19b-3p	Blood plasma from 7 UD patients, 7 BD patients, and 6 controls	-	MAPK1, PTEN, and PRKAA1	mTOR, FoxO, and the PI3-K/Akt signaling pathway	*p* = 0.0462	Candidate molecule analysis	MiR-19b-3p is a member of the miR-17/92 cluster, which controls lymphocyte growth, activation and proliferation.	[51]
miR-10b-5p	Skin biopsies of 3 control and 3 BP patient	Pluripotent Stem Cell-derived neurons	ANK3, BDNF, CAMK2G, DLGAP2, and NFASC	Axon guidance, Mapk, Ras, Hippo, Neurotrophin and Wnt signaling pathway	*p* < 0.05	Candidate molecule analysis(58 miRNAs)	-	[44]
miR-10b-3p			-					

**Table 7 ijms-22-05156-t007:** Altered expression of miRNAs following treatment with antipsychotic drugs.

miRNAs	Expression Pattern	Targets/Regulators	*p* Value	Function/Comments	Ref
hsa-miR-18a-5p	Up	-	*p* = 0.010761	These miRNAs were up-regulated in the Asenapine Group in this study. These findings suggest that candidate miRNAs might participate in the mechanism of function of both antipsychotics in bipolar mania.	[52]
hsa-miR-27a-3p			*p* = 0.000161		
hsa-miR-148b-3p			*p* = 0.005188		
hsa-miR-17-3p			*p* = 0.018034		
hsa-miR-106b-5p			*p* = 0.00445		
hsa-miR-106a-5p			*p* = 0.006898		
hsa-miR-20a-5p			*p* = 0.002247		
hsa-miR-17-5p			*p* =0.011219		
hsa-miR-19b-3p	Up	-	*p* = 0.013057	These miRNAs were up-regulated in the Asenapine Group. miR-19b, miR145, and miR-339, were formerly shown to be dysregulated in patients with autism spectrum disorder and with Alzheimer’s disease.	[52]
hsa-miR-145-5p			*p* = 0.029543		
hsa-miR-339-5p			*p* = 0.002185		
hsa-miR-15a-5p	Up	BDNF	*p* = 0.002422	hsa-miR-15a-5p was up-regulated in the Asenapine Group. miR-15a is reported to be involved in an interaction with brain-derived neurotrophic factor.	[52]
hsa-miR-30b-5p	Up	-	*p* = 0.015608	hsa-miR-30b-5p was up-regulated in the Asenapine Group. MiR-30b is associated with schizophrenia, a psychiatric disorder that has been shown to share common genetic roots with BD.	[52]
hsa-miR-210-3p	Up	-	*p* = 0.005157	hsa-miR-210-3p was up-regulated in the Asenapine Group. Overexpression of miR-210 induces angiogenesis and neurogenesis.	[52]
hsa-miR-92b-5p	Down	-	*p* = 0.04547	These miRNAs were down-regulated in the Asenapine Group in this study.	[52]
hsa-miR-1343-5p			*p* = 0.019721		
hsa-miR-664b-5p	Down	-	*p* = 0.035348	These miRNAs were down-regulated in the Risperidone Group in this study.	[52]
hsa-miR-6778-5p			*p* = 0.047124		
hsa-miR-146b-5p	Down	BDNF	*p* = 0.005919	hsa-miR-146b-5p was down-regulated in the Risperidone Group. miR-146b partakes in an interaction with brain-derived neurotrophic factor.	[52]

## Data Availability

The analyzed data sets generated during the study are available from the corresponding author on reasonable request.

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
