# Peer review of "A Comprehensive Review on the Role of Non-Coding RNAs in the Pathophysiology of Bipolar Disorder"

_ijms, 2021, doi:10.3390/ijms22105156_

Round 1
Reviewer 1 Report
The authors made an excellent literature review on the role of non-coding RNA in bipolar disorder pathology. The tables are a highlight. There are just a few minor things to be addressed:
- There is missing section of conclusions
- It would be interesting to add a few words of comparison on the role of ncRNA in other brain diseases such as stroke (PMID: 33327229), TBI (PMID: 29020847), cognitive impairment (PMID: 33094634) and others.
- A few errors are present like in this sequence: “are implicated in the complicated regulatory systems that regulateontrol expression of target genes”
Author Response
We thank Reviewer # 1 for the encouraging comments, which helped us to improve the quality of the manuscript. Please find the detailed point-by-point-response .
Thank you again for your valuable comments.
Comments and Suggestions for Authors
The authors made an excellent literature review on the role of non-coding RNA in bipolar
disorder pathology. The tables are a highlight. There are just a few minor things to be
addressed:
• There is missing section of conclusions
Response: We added a conclusion section (page 29, lines 68-70).
• It would be interesting to add a few words of comparison on the role of ncRNA in other brain
diseases such as stroke (PMID: 33327229), TBI (PMID: 29020847), cognitive impairment
(PMID: 33094634) and others.
Response: We added these references (page 28, lines 38-46).
• A few errors are present like in this sequence: “are implicated in the complicated regulatory
systems that regulateontrol expression of target genes”
Response: We corrected this sentence.
Reviewer 2 Report
The authors conducted a review of studies investigating the potential role of non-coding RNAs (miRNAs, lncRNAs and circRNAs) in the pathophysiology of bipolar disorder. The topic is interesting and of relevance.
The authors state they performed a "comprehensive search". If they performed a systematic review, all phases of the search should be described. Otherwise, the authors should clearly state this is a narrative review (and as such studies have been selected and discussed based on preference/choices from the authors).
Even in this case, since the authors state which databases they used for the search (and the fact that google scholar also includes grey literature besides peer-reviewed publications), I think the authors should add some information regarding which studies they considered to be eligible for inclusion and discussion (e.g. they discussed not only nonconding RNAs showing differential expression levels between patients with BD and controls, but also targets showing differential expression based on treatment and I think this should be specified).
The manuscript should be revised for:
- consistency in the use of acronyms (e.g. BD) once they are first used
- words in other fonts or in bold
Tables should be revised to be consistent as regard to
- the way in which sample size for different studies is reported
- the use or lack of spaces before and after the "=" or "<" characters
At page 2, line 59, please replace "regulateontrol" with "regulate control"
At page 2, line 60, please use an expression such as non-psychiatric controls rather then "normal subjects"
At page 2, line 88: "contribute" should be replaced with "contribute to"
At page 3, line 106: "dysregulation ... have been described" should be replaced with "dysregulation ... has been described"
At page 5, the following expression "in peripheral blood bipolar patients" should be replaced with something like "in peripheral blood of patients with BD"
Some miRNAs discussed in the text (e.g. miR-320a) seem to be missing from tables
Please check number pages in the last pages of the manuscript
In the last paragraph, "Finally, studies reported..." should be replaced with "Finally, studies reporting ..."
Author Response
We thank Reviewer # 2 for the encouraging comments, which helped us to improve the quality of the manuscript. Please find the detailed point-by-point-response .
Thank you again for your valuable comments.
The authors conducted a review of studies investigating the potential role of non-coding RNAs
(miRNAs, lncRNAs and circRNAs) in the pathophysiology of bipolar disorder. The topic is
interesting and of relevance.
The authors state they performed a "comprehensive search". If they performed a systematic
review, all phases of the search should be described. Otherwise, the authors should clearly state
this is a narrative review (and as such studies have been selected and discussed based on
preference/choices from the authors).
Even in this case, since the authors state which databases they used for the search (and the fact
that google scholar also includes grey literature besides peer-reviewed publications), I think
the authors should add some information regarding which studies they considered to be eligible
for inclusion and discussion (e.g. they discussed not only nonconding RNAs showing
differential expression levels between patients with BD and controls, but also targets showing
differential expression based on treatment and I think this should be specified).
Response: We stated that this study is a narrative review and discussed the mentioned points
(Page 2, lines 49-52).
The manuscript should be revised for:
• consistency in the use of acronyms (e.g. BD) once they are first used
• words in other fonts or in bold
Response: We corrected these points.
Tables should be revised to be consistent as regard to
• the way in which sample size for different studies is reported
• the use or lack of spaces before and after the "=" or "<" characters
Response: We checked Tables for this note.
At page 2, line 59, please replace "regulateontrol" with "regulate control"
At page 2, line 60, please use an expression such as non-psychiatric controls rather than
"normal subjects"
At page 2, line 88: "contribute" should be replaced with "contribute to"
At page 3, line 106: "dysregulation ... have been described" should be replaced with
"dysregulation ... has been described"
At page 5, the following expression "in peripheral blood bipolar patients" should be replaced
with something like "in peripheral blood of patients with BD"
Some miRNAs discussed in the text (e.g. miR-320a) seem to be missing from tables
Please check number pages in the last pages of the manuscript
In the last paragraph, "Finally, studies reported..." should be replaced with "Finally, studies
reporting ..."
Response: We have applied all of these points in the manuscript.